# Novel Leptin-Based Therapeutic Strategies to Limit Synaptic Dysfunction in Alzheimer’s Disease

**DOI:** 10.3390/ijms25137352

**Published:** 2024-07-04

**Authors:** Jenni Harvey

**Affiliations:** Department of Neuroscience, Ninewells Hospital and Medical School, University of Dundee, Dundee DD1 9SY, UK; j.z.harvey@dundee.ac.uk

**Keywords:** leptin, tau, hippocampus, synaptic plasticity, receptor trafficking, AMPA

## Abstract

Accumulation of hyper-phosphorylated tau and amyloid beta (Aβ) are key pathological hallmarks of Alzheimer’s disease (AD). Increasing evidence indicates that in the early pre-clinical stages of AD, phosphorylation and build-up of tau drives impairments in hippocampal excitatory synaptic function, which ultimately leads to cognitive deficits. Consequently, limiting tau-related synaptic abnormalities may have beneficial effects in AD. There is now significant evidence that the hippocampus is an important brain target for the endocrine hormone leptin and that leptin has pro-cognitive properties, as activation of synaptic leptin receptors markedly influences higher cognitive processes including learning and memory. Clinical studies have identified a link between the circulating leptin levels and the risk of AD, such that AD risk is elevated when leptin levels fall outwith the physiological range. This has fuelled interest in targeting the leptin system therapeutically. Accumulating evidence supports this possibility, as numerous studies have shown that leptin has protective effects in a variety of models of AD. Recent findings have demonstrated that leptin has beneficial effects in the preclinical stages of AD, as leptin prevents the early synaptic impairments driven by tau protein and amyloid β. Here we review recent findings that implicate the leptin system as a potential novel therapeutic target in AD.

## 1. Background

Alzheimer’s disease (AD) is a progressive neurodegenerative brain disorder, characterised by accumulation of plaques containing amyloid beta (Aβ) and neurofibrillary tangles comprised of hyper-phosphorylated tau (p-tau). Before plaques and tangles are formed, soluble toxic forms of tau and Aβ circulate in the brain and have been found to interfere with the normal functioning of synapses. Indeed, there is good evidence from cellular studies that exposure to either oligomeric tau [1] or Aβ [2,3] blocks the induction of long-term potentiation (LTP) at hippocampal CA1 synapses. LTP is a form of activity-dependent synaptic plasticity that is thought to be the cellular mechanism that is required for hippocampal-dependent learning and memory [4]. It is well established that an essential cellular event that contributes to activity-dependent synaptic plasticity is the trafficking of α-amino-3-hydroxy-5-methyl-4-isoxazolepropionic acid (AMPA) receptors into and out of synapses [5]. Several studies have demonstrated that acute treatment with Aβ influences AMPA receptor trafficking events such that Aβ leads to removal of AMPA receptors from hippocampal synapses [2]. More recent studies have demonstrated that treatment with oligomeric tau also promotes AMPA receptor endocytosis in hippocampal neurons [6]. Consequently, the ability of either tau or Aβ to block hippocampal LTP and promote internalisation of AMPA receptors is likely to underlie the cognitive impairments associated with AD. In addition to its acute actions, chronic exposure to Aβ drives loss of synapses, which ultimately leads to neuronal cell death.

## 2. Leptin and Its Receptors

It is now widely accepted that leptin, the obese (*ob*) gene product, is made and released from white adipose tissue, and the levels of leptin circulating in the plasma are directly proportional to body fat content [7]. Although leptin influences the functioning of various peripheral tissues, it can also readily enter the CNS, via transport across the blood–brain barrier. The arcuate nucleus and ventromedial nucleus are the primary targets for leptin in the hypothalamus, and these nuclei are essential sites for leptin to control food intake and body weight [8,9]. However, many studies have detected high levels of leptin receptor (LepR) expression in several other brain regions, including the hippocampus [10,11,12], where leptin has been found to regulate the cellular processes involved in learning and memory [13].

Leptin produces its biological effects via activating leptin receptors (LepRs), which are encoded by the LepR gene, and six splice variants have been identified (LepRa-f) [14]. The LepR isoforms are subdivided into two main types, short (LepR a,c,d,f) and long (LepRb) isoforms, based on their C-terminal domain lengths. LepRb is the main signalling competent form; however, the short LepR isoforms are reported to have some signalling capacity [15]. LepRe is dissimilar to the other isoforms as it lacks a membrane-spanning domain. However, LepRe retains a functional leptin binding site, and is hypothesised to act as a leptin carrier in the plasma.

Like other class 1 cytokine receptor family members, activation of LepRs initiates signalling via phosphorylation and activation of janus tyrosine kinases (JAKs) [16], which then stimulates various signalling pathways, including signal transducers and activators of transcription (STATs), phosphoinositide 3-kinase (PI3K) and ERK. LepR signalling is curtailed after stimulation of SOCS1-3, (suppressor of cytokine signalling), which then binds to phosphorylated JAK2, which triggers inhibition of JAK2 activity, thereby limiting further leptin action [17].

## 3. Leptin Regulates Hippocampal Excitatory Synaptic Function

The first reports that revealed possible hippocampal actions of leptin came from studies assessing the brain changes in leptin-deficient (*ob/ob*) compared to wild type mice. In these studies, Ahima et al. [18] determined that leptin deficiency was accompanied by significant developmental impairments in hippocampal and cortical brain regions; abnormalities that were readily reversed by administration of leptin. This study not only uncovered extrahypothalamic targets for the hormone leptin, but it was also the first to signal possible involvement of leptin in postnatal brain development. Later studies using in situ hybridisation and immunocytochemical techniques went on to confirm high levels of LepRb expression in the hippocampal formation [10,11]. But it was later studies carried out by Shanley and colleagues [12] that conclusively demonstrated synaptic expression of LepRs, as, significant co-localisation between LepR-positive staining and synapsin-1, a synaptic marker was detected in primary hippocampal neurons. Immunocytochemical analyses has also shown significant overlap between LepR expression and specific NMDA receptor subunits, which is also consistent with LepRs localised to hippocampal excitatory synapses [19].

Studies using electrophysiological techniques were the first to verify functional effects of leptin at hippocampal excitatory synapses. Thus, application of leptin to brain slices obtained from juvenile rats resulted in a transient reduction in excitatory synaptic transmission at the Schaffer-collateral (SC)-CA1 synaptic connection [20]. This effect of leptin was not species-dependent, as comparable effects of leptin were observed in mice at similar developmental stages [21]. Since these initial reports, numerous studies have characterised the developmental profile of leptin’s synaptic effects and revealed that the regulatory effects of leptin at hippocampal SC-CA1 synapses are highly dependent on age [22]. Thus, during early postnatal stages (P5-8), leptin evokes a novel form of long-term depression (LTD), whereas a persistent increase in synaptic transmission (leptin-induced LTP) is generated in response to leptin in adult hippocampal slices [22]. Irrespective of age, the ability of leptin to bi-directionally regulate excitatory synaptic function requires activation of NMDA receptors, as antagonism of NMDA receptors using D-AP5 blocks the effects of leptin on excitatory synaptic function at all ages [22,23]. Several, but not all, studies have observed that NMDA receptors composed of different GluN2 subunits are involved in the opposing forms of hippocampal synaptic plasticity, with GluN2B subunits identified as key for LTD induction, whereas LTP requires synaptic activation of GluN2A subunits [24,25]. Molecularly distinct NMDA receptors are also implicated in the divergent effects of leptin on excitatory synaptic efficacy, such that GluN2B-containing NMDA receptors are implicated in the depressant effects of leptin at early postnatal stages, whereas NMDA receptors comprised of GluN2A subunits are required for the chemical LTP induced by leptin in adulthood [22].

## 4. Regulation of Hippocampal Temporoammonic (TA)-CA1 Synapses by Leptin

In addition to being innervated via the classical tri-synaptic pathway which includes the SC input, hippocampal CA1 pyramidal neurons receive direct synaptic innervation via the TA pathway. The TA input starts in layer III of the entorhinal cortex, and it forms excitatory synaptic connections on distal CA1 dendrites located within *stratum-moleculare*. Unlike SC-CA1 synapses, TA-CA1 synapses are highly sensitive to dopamine, and dopamine is routinely used to verify stimulation of the TA input in electrophysiological recordings [26,27]. Recent evidence has found that, like SC-CA1 synapses, TA-CA1 synapses are also tightly controlled by the hormone leptin. Thus, in juvenile slices, exposure to leptin results in a novel form of chemical LTP at TA-CA1 synapses [26], which is contrary to the depressant effects of leptin observed at juvenile SC-CA1 synapses. Similarly, the synaptic effects of leptin at adult TA-CA1 synapses completely oppose those reported at SC-CA1 synapses, as treatment with leptin leads to induction of LTD in adult TA-CA1 synapses [28]. Although leptin has divergent effects on the dual synaptic inputs onto CA1 pyramidal neurons, there are some similarities in the cellular processes involved in the different forms of synaptic plasticity induced by leptin. Thus, NMDA receptor dependence, and, more specifically, involvement of selective NMDA receptor GluN2 subunits, is a key feature of leptin-induced synaptic plasticity at SC-CA1 and TA-CA1 synapses [22,26,28].

Collectively, the ability of leptin to induce persistent changes in the efficacy of synaptic transmission at TA-CA1 and SC-CA1 synapses has important implications for the functional effects of leptin, as it suggests that leptin has pro-cognitive properties. In support of this, numerous studies using a range of behavioural assays to monitor hippocampal-dependent learning and memory have observed improved performance in memory tasks in response to leptin [29,30,31,32].

## 5. Leptin Regulates Hippocampal AMPA Receptor Trafficking

It is widely accepted that delivery of AMPA receptors into and out of synaptic membranes is a crucial cellular event required for various forms of hippocampal synaptic plasticity and that LTP and LTD are associated with changes in the synaptic density of AMPA receptors [5]. There is good evidence that of the four AMPA receptor subunits that exist (GluA1-4), GluA1 subunits are vitally important for synaptic insertion of AMPA receptors during LTP [33,34], whereas GluA2 subunits are implicated in AMPA receptor endocytosis during LTD [35]. Additionally, several studies have observed transient switches in the subunit composition of synaptic AMPA receptors during the early phases of LTP [36,37]. In line with these studies, there is now good evidence that the persistent changes in synaptic strength induced by leptin involve delivery or removal of AMPA receptors from hippocampal synapses [23,28,38]. Moreover, a transient shift in the molecular composition of AMPA receptors from GluA2-containing to GluA2-lacking has been found to mediate leptin-induced LTP at SC-CA1 synapses, as evidenced by the transient increase in the rectification properties of synaptic AMPA receptors [23]. Parallel studies in hippocampal neurons supported these findings, as treatment with leptin increased the surface and synaptic expression of GluA1 subunits [23].

## 6. Leptin and Alzheimer’s Disease

It is known that as humans age the functioning of metabolic systems slowly deteriorates, which impacts not only peripheral tissues but also overall brain health, as many brain regions are important targets for metabolic hormones. Mounting evidence indicates that during the ageing process brain sensitivity to leptin is reduced [28,39] and that decreased leptin uptake into the brain may contribute to this altered leptin sensitivity [40]. Ageing is also linked to an increased likelihood of developing leptin resistance, due to mid-life driven increases in body fat, which subsequently lead to elevations in the plasma levels of leptin [41]. Furthermore, evidence from clinical studies confirms that an obese phenotype is associated with a higher risk of developing neurodegenerative disease, including amyotrophic lateral sclerosis (ALS) [42,43], Parkinson’s disease (PD) [44] as well as Alzheimer’s disease (AD) [45,46].

The association between body adiposity and neurodegenerative disease risk suggests a pivotal role of adipokines like leptin. In support of this, prospective clinical studies have identified a link between plasma leptin levels and future risk of AD [47,48]. Moreover, uncharacteristically low circulating levels of leptin are common in AD patients, suggesting that malfunctions in the leptin system are linked to the disease process [49]. Lower than normal leptin levels also manifest in various rodent models of AD [30,50], indicating that parallel alterations in key metabolic hormones also occurs in other mammals. These observations suggest that enhancing the brain leptin levels could have restorative, therapeutic effects in AD. However, some clinical studies have found no alterations in leptin levels in AD patients [51,52] but have detected altered leptin driven signalling in the later stages of the disease [53]. Similarly, in the 5XFAD AD model, impaired leptin receptor signalling has been observed in late-stage AD. Consequently, it is feasible that leptin may only be a viable treatment option in the early stages of disease, when synaptic abnormalities occur.

## 7. Leptin Has Neuroprotective Actions

Since the initial studies performed by Ahima and colleagues [18] that identified a direct correlation between neuronal survival and loss of leptin, numerous studies have reported that leptin has widespread neuroprotective actions, with leptin not only improving neuronal viability but also protecting neurons against various toxic stimuli [54,55]. Activation of classical pro-survival pathways, such as PI3K-Akt and JAK-STAT3 signalling, as well as improved mitochondrial function have been implicated in the protective mechanisms of leptin [54,56]. The pro-survival actions of leptin extend to several other forms of CNS-driven disease, as there is good evidence to support the protective effects of leptin in various models of ischaemic stroke [57,58,59].

More recent evidence points to a protective role for leptin in various neurodegenerative diseases, including AD [50,60,61]. One key pathological trait of AD is the formation of plaques containing toxic forms of amyloid beta (Aβ), which accumulate due to sequential cleavage of amyloid precursor protein (APP) by β- and γ secretase. Increasing evidence indicates that leptin limits this process by reducing expression of both secretases and via directly inhibiting β-secretase activity [62,63]. Studies by Greco et al. [64,65] observed that treatment of neuronal cells with leptin promoted Aβ uptake, thereby attenuating the extracellular Aβ levels. Moreover, in APP_swe_-expressing mice or APP/PS1 mice, exposure to leptin resulted in decreased brain amyloid load [60,66]. Collectively, these findings indicate that treatment with leptin reduces amyloid burden in AD.

In addition to Aβ, neurofibrillary tangles containing hyper-phosphorylated tau also build up in the brain during AD. Phosphorylation of tau protein is regulated by glycogen synthase kinase 3β (GSK3β), a serine/threonine kinase that is inhibited downstream of PI3K. Recent studies in PC12 neuronal cells and rodent models of AD have shown that treatment with leptin reduces tau phosphorylation by stimulating PI3K, which limits activation of GSK3β [64,67]. In accordance with leptin restricting tau phosphorylation, elevated brain levels of phosphorylated tau (p-tau) are observed in rodents with insensitivity or resistance to leptin [68,69].

## 8. Leptin Limits the Synapto-Toxic Effects of Aβ and Tau

Numerous lines of evidence indicate that oligomeric Aβ has detrimental effects at synapses in the early pre-clinical stages of AD. Thus, studies using electrophysiological approaches have shown that treatment with oligomeric Aβ blocks the induction of hippocampal LTP in acute brain slices [2,3]. Oligomeric Aβ also facilitates the induction of LTD, such that sub-threshold low frequency stimulation paradigms that are unable to induce LTD in control conditions can induce robust LTD in slices treated with Aβ [3]. Recent studies indicate that in hippocampal slices exposed to leptin, the aberrant effects of Aβ on both forms of synaptic plasticity are prevented [68]. There is also good evidence that acute exposure to Aβ promotes removal of GluA1-containing AMPA receptors from hippocampal synapses [2]; an effect that is also blocked by leptin [68]. Pharmacological studies examining the signalling pathways mediating the protective actions of leptin have identified a key role for PI3K, as inhibition of PI3K activity blocked the ability of leptin to prevent Aβ-driven internalisation of GluA2-lacking AMPA receptors and inhibition of LTP induction [68]. Although the leptin-dependent protective pathways activated downstream of PI3K have yet to be determined, the serine threonine kinase, GSK3β, is the likeliest potential target, as previous studies have uncovered that inhibition of GSK3β mimics and occludes the effects of leptin [65].

It is known that the microtubule associated protein (MAP) tau is primarily expressed in axonal regions, where it controls the rate of microtubule assembly and stabilisation [70]. In healthy tissue, there is limited expression of tau in somato-dendritic regions [71]. However, in AD, there is aberrant targeting and accumulation of tau at synapses, which is associated with synapse loss and degeneration. Increasing evidence indicates that hyperphosphorylation of tau is a key factor that drives its synapto-toxic effects. Phosphorylation of tau not only restricts its interaction with microtubules, but it also drives movement of tau from axons to synapses, which subsequently leads to AMPA receptor endocytosis and deficits in excitatory synaptic transmission [72]. The tau-linked changes in hippocampal synaptic function are thought to be early abnormal synaptic changes that drive tau-related pathology and are a key contributory factor in the cognitive impairments observed in AD.

It is known that exposure to Aβ oligomers can promote an increase in tau phosphorylation [73] via a process that requires activation of GSK-3β, the key kinase involved in tau phosphorylation [74,75]. Moreover, chronic treatment with Aβ can be used to model tau-related synaptic dysfunction, as this has been shown to drive tau phosphorylation and its targeting to synapses [72,76]. In line with these studies, recent evidence from immunocytochemical studies has found that chronic treatment of hippocampal neurons with Aβ resulted in movement of tau from axons to dendrites [6]. Moreover, in hippocampal neurons treated with leptin, the ability of Aβ to promote an increase in the dendritic and synaptic levels of tau was prevented [6]. Increased expression of phosphorylated tau at serine 396 (Ser396) is well documented in AD, and there is good correlation between phosphorylation of the tau Ser396 site and neuronal cell death [77,78]. Studies using a phospho-(Ser396) specific antibody showed that movement of tau to synapses is enhanced when tau Ser396 becomes phosphorylated [6]. Moreover, exposure to leptin not only blocked phosphorylation of tau at Ser396, but it also prevented tau insertion into synapses. The ability of leptin to limit trafficking of tau to hippocampal synapses is likely to involve PI3-kinase dependent inhibition of GSK-3β, as pharmacological inhibition of PI3-kinase blocked leptin action, whereas addition of GSK-3β inhibitors mimicked and occluded the protective effects of leptin. The ability of leptin to limit tau phosphorylation via inhibiting GSK-3β is in accordance with previous studies performed in SH-SY5Y cells [65], which demonstrated that inhibition of GSK3β mediates leptin-dependent inhibition of tau phosphorylation in this neuronal cell line.

Our recent studies observed that synaptic insertion of tau resulted in the internalisation and removal of AMPA receptors from hippocampal synapses, as increased dendritic expression of tau was associated with reductions in both the surface and synaptic expression of the AMPA receptor subunit, GluA1. Treatment with leptin also prevented the tau-related attenuation in surface GluA1 expression in hippocampal neurons [6]. In support of tau driving aberrant changes in AMPA receptor trafficking events, exposure of hippocampal neurons to oligomeric tau also significantly reduced surface GluA1 expression, suggesting that tau directly promotes AMPA receptor endocytosis (see Figure 1). This effect was also prevented by prior exposure to leptin, indicating that leptin limits tau-driven internalisation of AMPA receptors [6]. Several studies have shown that in acute hippocampal slices treated with oligomeric tau, high frequency stimulation failed to induce LTP at hippocampal synapses [1]. In line with this, treatment with oligomeric tau blocked the induction of LTP at hippocampal SC-CA1 synapses, and treatment with leptin blocked this aberrant effect of tau on synaptic plasticity [6]. Collectively, these findings indicate that leptin limits tau phosphorylation and subsequent insertion of tau into hippocampal synapses. Furthermore, leptin prevents the direct effects of tau on the functionality of hippocampal synapses by reducing tau-driven AMPA receptor endocytosis and blocking the induction of hippocampal synaptic plasticity. As modification of tau and its accumulation at synapses strongly correlates with cognitive impairments in AD, the ability of leptin to limit tau phosphorylation and its targeting to synapses has important implications for the neuroprotective actions of leptin in CNS disorders like AD.

It is well established that activity-dependent hippocampal synaptic plasticity is a cellular correlate of learning and memory [4]. Moreover, the aberrant effects of Aβ and tau at hippocampal synapses result in impaired memory function, which closely mirrors the memory deficits observed in AD. In line with the observed protective effects of leptin at hippocampal synapses, there is now good evidence that leptin also alleviates AD-driven cognitive impairments reported in various rodent models of AD. Indeed, treatment of SAMP8 mice with leptin gives rise to enhanced performance in hippocampal memory tasks [30]. The ability of CRND8 mice to perform novel object recognition tasks to measure episodic memory is also improved by leptin [64]. Administration of leptin also attenuates Aβ_1–42_-driven impairments in spatial memory in wild type mice and AD-driven memory impairments in APP/PS1 mice [66]. In CRND8 transgenic mice, chronic administration of leptin via intracerebroventricular injection over an 8-week period also reduced overall hippocampal Aβ burden and improved performance in novel object recognition and contextual and cued fear conditioning tasks [50]. Injection of Aβ_1–42_ into the lateral ventricle of wild-type rats results in impairments in hippocampal spatial memory as well as suppression of hippocampal LTP, and these effects are alleviated by administration of leptin [61]. Collectively, these findings indicate that treatment with leptin improves memory in various AD models.

## 9. Leptin-Based Molecules as Therapeutics?

Leptin is already licenced for use as a drug to treat obesity, and, in clinical studies, treatment with leptin is very effective as it dramatically reduces body weight in individuals with morbid obesity attributed to recessive *ob* gene mutations. In these studies, Matochik et al. [79] discovered that administration of leptin not only reversed the obese phenotype but also gave rise to pro-cognitive effects, as evidenced by increased grey matter volume in functional brain imaging. In a comparable study, significant cognitive improvements were observed in response to leptin in a 5-year-old child with congenital leptin deficiency [80], thereby indicating that leptin-based therapies readily cross the blood–brain barrier and can penetrate key brain regions involved in cognitive processes, such as the hippocampus. Although numerous studies indicate that leptin has positive effects in pre-clinical AD models, clinical trials using leptin have yet to be carried out in individuals with AD, and, consequently, the clinical efficacy of leptin remains to be determined. However, there are likely to be caveats to using leptin therapeutically, as not every individual suffering from AD would benefit from leptin treatment. In particular, leptin-based molecules are unlikely to be effective in treating AD patients who display resistance to leptin. Nevertheless, boosting leptin levels could be beneficial for those with low plasma leptin levels. In support of this possibility, there is now good evidence from clinical studies that recombinant human methionyl leptin (metreleptin), which is used to treat lipodystrophies, characterised by very low leptin levels due to loss of adipose tissue, has positive cognitive benefits [81,82]. Moreover, administration of metreleptin in individuals with low leptin levels due to anorexia nervosa also leads to significant improvements in cognitive function [83]. In view of these findings, it is likely that a precision medicine-based approach may be needed to identify suitable AD patients for leptin-based therapies.

One significant advance that has been made is in the development and verification of small leptin-based peptides as possible therapies for AD. Previous work identified leptin-derived fragments with anti-obesity actions [84]. However, recent evidence indicates that one of the leptin fragments, leptin_116–130_, fully recapitulates the pro-cognitive and neuroprotective actions of whole leptin in various models of AD [31]. More recent studies by Doherty and colleagues [29] have extended this further by identifying that four smaller 6 amino acid hexamers (leptin_116–121_, leptin_117–122_, leptin_118–123_ and leptin_120–125_) mirror the beneficial effects of leptin. Thus, the hexamers were found to promote AMPA receptor trafficking to synapses and to enhance the magnitude of hippocampal LTP. In behavioural assays, peripheral administration of one hexamer (leptin_117–122_) resulted in improved performance in novel object preference tasks that model human episodic-like memory. In cellular models of AD, treatment with the bioactive hexamers attenuated the acute aberrant effects of Aβ on hippocampal synapses as well as limiting neuronal cell death and tau phosphorylation induced by chronic exposure to Aβ [29]. Interestingly, myristic acid conjugated leptin peptides have also been developed that are biologically active and have therapeutic potential [85]. Thus, a peptide known as MA-[D-Leu-4]-OB3 is reported to have beneficial effects in diabetic models and AD-like cognitive impairment [86,87]. Indeed, diabetic models treated with MA-[D-Leu-4]-OB3 display restored glycaemic control as well as improvements in episodic memory [86], with the improvements in cognitive function associated with enhanced insulin sensitivity [87]. Collectively, these findings support the notion that leptin-based small molecules are feasible therapeutic targets for treating AD. Although these findings offer some hope, there is still significant experimental evidence required before leptin-based molecules can be advanced for use in clinical trials.

In conclusion, there is now significant evidence that the adipokine, leptin, protects against the detrimental synapto-toxic effects of Aβ [29,31]. More recent evidence indicates that the protective actions of leptin extend to tau protein, as the aberrant effects of tau at hippocampal synapses are also limited by leptin. Collectively, these findings add to the growing evidence that supports the notion that boosting brain levels of leptin not only counteracts Aβ- and tau-driven synaptic dysfunction but also that targeting the leptin system may have therapeutic value in treating AD. Overall, these findings have implications for neurodegenerative disorders like AD. Potential limitations in the use of leptin therapeutically in AD patients with leptin resistance has possibly hampered progress in its clinical application. However, further clinical assessment of the therapeutic value of leptin-based molecules is needed if effective treatment options are to be developed for patients in the future.

## Figures and Tables

**Figure 1 ijms-25-07352-f001:**
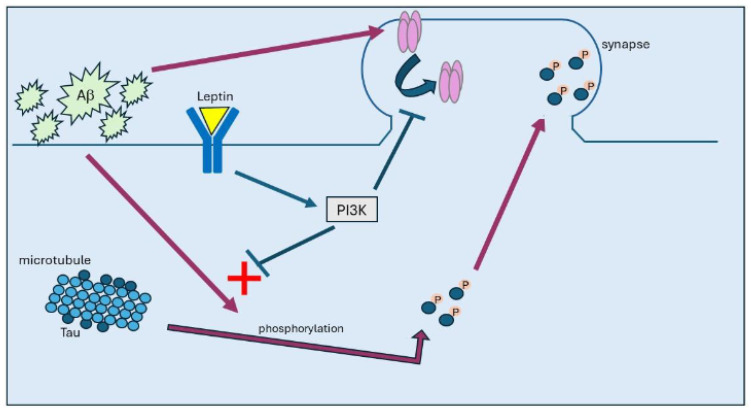
In AD, build-up of Aβ promotes hyperphosphorylation of tau, which attenuates its affinity to bind to microtubules. Once hyperphosphorylated, tau then moves from axonal regions to synapses, where it accumulates, leading to internalisation of AMPA receptors. Treatment with leptin results in activation of PI3K and subsequent inhibition of GSK3β, which prevents tau phosphorylation and its subsequent trafficking to synapses.

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
