# Peer review of "Novel Leptin-Based Therapeutic Strategies to Limit Synaptic Dysfunction in Alzheimer’s Disease"

_ijms, 2024, doi:10.3390/ijms25137352_

Round 1
Reviewer 1 Report
Comments and Suggestions for Authors
The review article entitled “Novel therapeutic strategies to limit tau-related synaptic function in Alzheimer’s disease” by Jenni Harvey focuses on the memory benefit of leptin levels in Alzheimer’s disease (AD) and its association to tau related pathology. The main objective is to inform readers about the potential of leptin as a potential therapeutic target for AD treatment. The article comprehensively covers major studies on leptin. It references significant research milestones on leptin and its roles in diseases, using clear and easily understanding language. The topic is highly relevant and will encourage readers to explore novel treatments for AD. However, there are few issues that need to be addressed before it can be accepted for publication:
1. Line 30 should be revised to state “soluble toxic forms of Aβ” instead of “soluble forms of Aβ” since monomeric form should not be considered toxic.
2. Please spell out AMPA and use the abbreviation throughout the entire manuscript
3. In section #6., the author references both leptin resistance and its therapeutic benefit. Please provide a clear explanation of the conflicting results by discussing the underlying mechanisms.
4. In the conclusion section, the author mentions the 2013 publication on leptin’s protection against Aβ toxicity. Please explain why there has been limited reporting or progress in clinical applications since then.
Author Response
I would like to thank the reviewers for their constructive comments on the manuscript. The response to the queries raised are detailed below.
Reviewer 1:
Line 30 should be revised to state “soluble toxic forms of Aβ” instead of “soluble forms of Aβ” since monomeric form should not be considered toxic.
This has now been amended.
- Please spell out AMPA and use the abbreviation throughout the entire manuscript.
This has now been amended.
- In section #6., the author references both leptin resistance and its therapeutic benefit. Please provide a clear explanation of the conflicting results by discussing the underlying mechanisms.
This has now been fully discussed in the manuscript.
- In the conclusion section, the author mentions the 2013 publication on leptin’s protection against Aβ toxicity. Please explain why there has been limited reporting or progress in clinical applications since then.
This has now been amended in the conclusion section.
Reviewer 2 Report
Comments and Suggestions for Authors
The review by Dr Harvey is relevant and interesting but it represents a very brief overview of the subject. I would suggest developing the subject further. For example, given that the main purpose of the review was to advocate for a therapeutic strategy based on leptin, it would have been relevant to expand on the preclinical application of leptin in Alzheimer's models and the findings so far. I suggest including some references including but not limited to: Pratap and Holsinger https://doi.org/10.3390/ph13110401; Scott et al. doi: 10.1002/cne.22025. Maioli et al. doi: 10.1111/acel.12281. Pérez-González et al 10.3233/JAD-2011-102070.
However, my main comment relates to the title” Novel therapeutic strategies to limit tau-related synaptic function in Alzheimer’s disease.” This is highly misleading. It should mention leptin. Same applies for the abstract. In the text, the author discusses both amyloid and tau effects and the implication of leptin on both pathologies, which is appreciated. This should be reflected in both the title and abstract.
In addition, human studies with leptin have shown mixed results with many studies indicating no change in plasma levels of leptin in populations with or at risk of AD and/or no correlation with cognitive status. This combined with a resistance or insensitivity to leptin as briefly mentioned in this review. In such panorama, what would be the benefit of a therapeutic application of leptin? Some papers worth to be cited: Oania et al. doi: 10.1093/ageing/afu160.; Teunissen et al. doi: 10.3233/JAD-141503.; Baranowska-Bik et al. doi: 10.1016/j.npep.2015.05.006.; Tian et al. doi: 10.3233/JAD-220447.
Author Response
Reviewer 2:
given that the main purpose of the review was to advocate for a therapeutic strategy based on leptin, it would have been relevant to expand on the preclinical application of leptin in Alzheimer's models and the findings so far.
Additional information on the preclinical actions of leptin in AD models has now been included.
However, my main comment relates to the title” Novel therapeutic strategies to limit tau-related synaptic function in Alzheimer’s disease.” This is highly misleading. It should mention leptin. Same applies for the abstract. In the text, the author discusses both amyloid and tau effects and the implication of leptin on both pathologies, which is appreciated. This should be reflected in both the title and abstract.
The title has now been amended to reflect this.
In addition, human studies with leptin have shown mixed results with many studies indicating no change in plasma levels of leptin in populations with or at risk of AD and/or no correlation with cognitive status. This combined with a resistance or insensitivity to leptin as briefly mentioned in this review. In such panorama, what would be the benefit of a therapeutic application of leptin? Some papers worth to be cited: Oania et al. doi: 10.1093/ageing/afu160.; Teunissen et al. doi: 10.3233/JAD-141503.; Baranowsk
This has now been discussed further in the manuscript, and relevant references added.
Round 2
Reviewer 2 Report
Comments and Suggestions for Authors
It is appreciated that the title was changed to better reflect the contents.
However, although some information has been added (2 references added), it is still lacking an in-depth overview of the animal and human work related to leptin levels and therapy.
Author Response
I would like to thank the reviewer for their additional comments. the rebuttal to this is detailed bel;ow.
Reviewer 2:
However, although some information has been added (2 references added), it is still lacking an in-depth overview of the animal and human work related to leptin levels and therapy.
Significant additional information had already been added to the manuscript which detailed the reported beneficial effects of leptin in AD models (page 10). In addition we had already added additional information on the actions of leptin in humans. We have now extended this section (on page 11) to include new information on metreleptin.